# Plasmon-Enhanced Blue-Light Emission of Stable Perovskite Quantum Dot Membranes

**DOI:** 10.3390/nano9050770

**Published:** 2019-05-19

**Authors:** Kai Gu, Hongshang Peng, Siwei Hua, Yusong Qu, Di Yang

**Affiliations:** 1College of Science, Minzu University of China, Beijing 10081, China; kai_gu94@163.com (K.G.); hshpeng@bjtu.edu.cn (H.P.); 17301037@muc.edu.cn (S.H.); quyusong123@126.com (Y.Q.); 2Key Laboratory of Luminescence and Optical Information, Ministry of Education, Institute of Optoelectronic Technology, Beijing Jiaotong University, Beijing 100044, China

**Keywords:** perovskite quantum dots, plasmon-enhance fluorescence, electrospinning

## Abstract

A series of stable and color-tunable MAPbBr_3−x_Cl_x_ quantum dot membranes were fabricated via a cost-efficient high-throughput technology. MAPbBr_3−x_Cl_x_ quantum dots grown in-situ in polyvinylidene fluoride electrospun nanofibers exhibit extraordinary stability. As polyvinylidene fluoride can prevent the molecular group MA^+^ from aggregating, MAPbBr_3−x_Cl_x_ quantum dots are several nanometers and monodisperse in polyvinylidene fluoride fiber. As-prepared MAPbBr_3−x_Cl_x_ quantum dot membranes exhibit the variable luminous color by controlling the Cl^−^ content of MAPbBr_3−x_Cl_x_ quantum dots. To improve blue-light emission efficiency, we successfully introduced Ag nanoparticle nanofibers into MAPbBr_1.2_Cl_1.8_ quantum dot membranes via layer-by-layer electrospinning and obtained ~4.8 folds fluorescence enhancement for one unit. Furthermore, the originality explanation for the fluorescence enhancement of MAPbBr_3−x_Cl_x_ quantum dots is proposed based on simulating optical field distribution of the research system.

## 1. Introduction

Hybrid organic-inorganic perovskites have made remarkable achievements in the application of photovoltaic devices [1], but beyond that they have garnered considerable attention in light-emitting applications due to their low-cost, facile preparations [2], flexible tunability of emission light [3], extraordinary charge-transport properties [4], and promisingly ultrahigh photoluminescence [5,6]. However, hybrid organic-inorganic perovskites suffer from environmental instability and are frustrated in light-emitting applications [7]. As an important class of hybrid organic-inorganic perovskites, MAPbX_3_ (MA = CH_3_NH_3_^+^, X = Cl, Br, I) has been widely believed to possess an inoxidizability ground state in oxygen atmospheres, and more importantly, when MAPbX_3_ quantum dots (QDs) are exposed to oxygen atmospheres, their photoluminescence (PL) intensities are enhanced [8]. Recently, MAPbX_3_ QDs has become a promising candidate in breaking through the short-term stability and gain important positions in many light-emitting applications such as light-emitting diodes (LEDs), lasers and light-emitting transistors [9,10].

Among the family of MAPbX_3_ QDs, many reports have mainly focused on MAPbBr_3_ QDs, which is a green emitter with excellent photoluminescence quantum yield (PLQY, ~90%) [11,12]. By doping MAPbBr_3_ with chlorine, stoichiometric MAPbBr_3−x_Cl_x_ can be synthesized. The bandgap of MAPbBr_3−x_Cl_x_ widens as Cl^−^ proportion increases and the emission wavelength can be tuned continuously from 400 to 520 nm by adjusting the ratio of Cl^−^ to Br^−^ of MAPbBr_3−x_Cl_x_ [13]. Unfortunately, with the increase of bandgap width of MAPbBr_3−x_Cl_x_, more defect energy levels located in the bandgap are likely to serve as deep level recombination centers, and MAPbBr_3−x_Cl_x_ QDs generally suffer from low PLQY [14]. One major source of such defects is associated with dangling bonds on the surface of MAPbBr_3−x_Cl_x_ QDs and surface passivation is an effective approach to remove these and achieve efficient photoluminescence. The polymer matrices with compact molecular chains were reported to be effective in passivating perovskite quantum dots (PQDs) and the polymer-encapsulated PQDs exhibit high environment stability especially water resistance [15].

A cost-efficient high-throughput strategy for fabricating polymer-encapsulated PQDs has recently become one research hotspot due to its necessity and importance to the future industrialization of these materials. Spin-coating is usually used to fabricate large-area membranes. However, in a spin-coating process, there is not only a serious waste of materials, but it is also difficult to control the uniformity of large-area membranes [16]. These daunting problems disappear when electrospinning is used to prepare large-area membranes. More interestingly, electrospinning is quite capable of fabricating various functional nanofibers, thus leading to plentiful and novel performances from the large-area membranes made of the nanofibers. Recently, polymer-encapsulated PQDs prepared by electrospinning have attracted great attention [17,18,19]. For instance, Tsai et al. [20] used polyacrylonitrile and perovskite precursor as shell and core materials, respectively, and prepared core-shell coaxial nanofibers through one-step coaxial electrospinning method. They revealed that these nanofibers were endowed with high ambient stability and a high resistance to water. Kuo et al. [21] reported a novel electrospinning strategy for incorporating CsPbX_3_ (X = Cl, Br, I) QDs into poly (styrene-butadiene-styrene). The protected PQDs continued to emit bright fluorescence for over 1 h when immersed in water.

Even more remarkably, electrospinning could knit nanofibers embodying plentiful plasmonic nanoparticles and PQDs together in the most optimal way, which ensures extraordinary fluorescence enhancement. Although plasmon-enhanced fluorescence has been widely employed to boost the luminescence yields of fluorescence dyes and QDs [22,23,24,25,26], few studies have focused on the fluorescence enhancement system that plasmonic nanofibers are interwoven with QD nanofibers. Therefore, there is a lack of thorough understanding of fluorescence enhancement and the interaction mechanism between the two types of nanofibers for this system, which is different from the traditional interactions between particles and/or films.

Herein, we successfully prepared uniform membranes of polymer-encapsulated MAPbBr_3−x_Cl_x_ QDs via a straightforward uniaxial electrospinning method. Polyvinylidene fluoride (PVDF) was selected to endow PQDs high stability due to its hydrophobic characteristics, semi-crystalline and strong ability to restrict PQDs growth [27]. The as-fabricated PVDF-encapsulated PQDs (PVDF-PQDs) membranes exhibit color-tunability by controlling the ratio of Cl^−^ to Br^−^ in MAPbBr_3−x_Cl_x_. On this basis, the MAPbBr_3−x_Cl_x_, which the PL spectrum coincides well with plasmonic resonance band of silver nanoparticles (NPs), were selected for Ag enhanced fluorescence experiments. Ag nanofibers were introduced into PQD nanofiber membrane via layer-by-layer electrospinning method and Ag nanofibers of each layer interweaved with PQD nanofibers of the adjacent layer. We defined PQDs-layer/Ag NPs-layer/ PQDs-layer as an analytical periodic unit and proved that one unit possesses a ~4.8 folds fluorescence enhancement, which is a significant improvement compared with ~1.7 folds fluorescence enhancement for CsPbBr_3_ QDs and ~4.0 folds fluorescence enhancement for MAPbBr_3_ QDs reported elsewhere [23,26]. Furthermore, we interpreted the plasmonic enhancement mechanism of this system by simulating the light field distribution of Ag nanofibers.

## 2. Materials and Methods

### 2.1. Materials

PbCl_2_ (99.99%), MABr (MA = CH3NH_3_^+^, 99.5%), PbBr_2_ (99.99%) and MACl (99.5%) were purchased from Xi’an Polymer Light Technology Co. Ltd., Xi’an, China. *N*,*N*-dimethylformamide (DMF, 99.5%) and Poly(*N*-vinylpyrrolidone) (PVP) with molecular weight (Mw) ~130,000 were obtained from Aladdin (Shanghai, China). Another PVP with Mw ≈ 40,000 and PVDF pellets (Mw ≈ 275,000) were gotten from Sigma-Aldrich (Shanghai, China). AgNO_3_ (99.98%), ethylene glycol and acetone were purchased from Beijing Chemical Reagent Co. Ltd., Beijing, China. All the chemicals were used without further purification.

### 2.2. Preparation of Perovskite Precursor Solutions

A series of precursor solutions of PVDF-encapsulated MAPbBr_3−x_Cl_x_ (PVDF-MAPbBr_3−x_Cl_x_) were prepared via a simple stirring process. First, PVDF of 2 g was poured into 10 mL DMF and stirred for 1.5 h at 70 °C to dissolve it completely. After the PVDF solution slowly cooling down to room temperature, the precursor mixed by MABr and PbCl_2_ was added to the PVDF solution during stirring and the precursor solution was obtained after continuous stirring at room temperature for ~2 h. The molar concentrations of MABr and PbCl_2_ in the precursor solution were 0.13 and 0.10 M, respectively. Similarly, to obtain MAPbBr_3_ and MAPbCl_3_, a DMF solution containing both MABr (0.1 M) and PbBr_2_ (0.1 M), as well as both MACl (0.1 M) and PbCl_2_ (0.1 M) was prepared according to the above process, respectively.

### 2.3. Synthesis of Ag NPs

In preparing Ag NPs, a known quantity of AgNO_3_ was added to 30 mL ethylene glycol solution of PVP (Mw ≈ 40,000). The mixture was stirred at 110 °C for 1 h in an oil bath. Finally, Ag nanoparticles were collected by centrifugation. It was found that with the benefit of acetone precipitator, the high yield of Ag nanoparticles was obtained by centrifuging the as-prepared solution at 10,000 r/min.

### 2.4. Preparation of Ag NPs Electrospinning Solutions

First, PVP (MW ≈ 130,000) was dissolved into DMF. In a typical dissolution process, 1 g PVP was first dissolved in 10 mL DMF and stirred for 1 h at the room temperature. Subsequently, the as-produced Ag NPs were added into the PVP solution to prepare a series of electrospinning solutions with Ag NPs concentrations of 0, 0.067, 0.118, 0.236 and 0.324 M, respectively.

### 2.5. Preparation of Polymer-Encapsulated Perovskite Electrospinning Membrane with and without Containing Plasmonic Nanoparticle Nanofibers

Based on the as-prepared solutions, that is, MAPbBr_3−x_Cl_x_ precursor solutions or Ag NP solutions, we fabricated MAPbBr_3−x_Cl_x_ membranes and Ag-enhanced MAPbBr_1.2_Cl_1.8_ membranes by electrospinning, respectively. Herein, needles with diameter of 1.25 mm were utilized, the distance between the spinning needle tip and the deposited substrate was 10 cm, and the working voltage was set to 19 kV. The flow rate of different kinds of solutions was controlled at 0.2 mL/h. To facilitate the comparison of photoluminescence, we prepared different MAPbBr_3−x_Cl_x_ membranes on the same size substrates with the same deposition time (60 min). Similarly, the membrane of plasmon-enhanced fluorescence of PQDs was fabricated on the substrate of the same size. A layer of PVP-encapsulated Ag NPs (PVP-Ag) which was deposited for 1 min was added at the centrality of the MAPbBr_1.2_Cl_1.8_ membranes and the deposited time for both the lower and the upper MAPbBr_1.2_Cl_1.8_ layers were 30 min.

### 2.6. Characterizations

Photoluminescence spectra were measured via using OmniFluo fluorescence spectrometer (Zolix, Beijing, China). Ultraviolet-visible (UV-vis) absorption spectra were taken on a UV-3600 Plus spectrophotometer (SHIMADZU CO., Ltd., Hong Kong, China). X-ray diffraction (XRD) measurements were measured on a Bruker/D8 FOCUS X-ray diffractometer (Billerica, MA, USA) with a Cu Kα radiation source (wavelength at 1.5405 Å). Transmission electron microscope (TEM) and high-resolution TEM (HRTEM) images were recorded with a Technai F20 transmission electron microscope (FEI, Hillsboro, OR, USA). Scanning electron microscopy (SEM) images were taken on a S-4800 SEM microscope (Hitachi, Ltd., Tokyo, Japan).

### 2.7. Theoretical Modeling

Previously, we calculated optical field distributions of plasmonic nanoparticles (Ag and Au) using the finite element method, and derived the plasmon absorption spectra, which can completely reproduce the measured ones [28,29]. On the basis of this computing technology, simulation of the optical field distribution of the PVP-Ag nanofiber was performed. Theoretical modeling was based on SEM and TEM measurements of as-prepared samples. The dielectric constants of PVP, PVDF and Ag used here were taken from earlier studies [30,31,32]. The incident electric field, *E*_y_ = exp(*jk*_x_*x*), propagated along the x-axis with polarization in the y-direction. The computational domain was artificially truncated by applying scattering boundary conditions to the upper and lower boundaries, while putting periodic boundary conditions on the left and right boundaries.

## 3. Results and Discussion

### 3.1. PVDF-Encapsulated MAPbBr_3−x_Cl_x_ QDs

A series of uniform large area PVDF-encapsulated MAPbBr_3−x_Cl_x_ (PVDF-MAPbBr_3−x_Cl_x_) with x = 0, 1.8, and 3 membranes were fabricated by electrospinning. As constituent elements of these membranes, the nanofibers are highly uniform in dimensions and their surfaces are smooth, and, more importantly, PQDs are completely encapsulated inside the nanofibers, thus gaining excellent surface passivation. For example, Figure 1a,b show SEM and TEM images of PVDF-MAPbBr_1.2_Cl_1.8_ nanofibers. Obviously, almost no PQDs are embedded on the surface of the nanofibers, which makes the surface of the nanofibers smooth (Further support in Appendix A). Notably, the highly efficient passivation of PQDs fabricated by such a simple method is equivalent to that of the perovskite/polymer core/shell nanostructure, whereas this method avoids the tedious processes of synthesizing core/shell nanostructures. The average diameter of as-prepared nanofibers was 68.1 nm and a histogram of the dimension distribution is shown in Appendix A.

A HRTEM image of PVDF-MAPbBr_1.2_Cl_1.8_ nanofibers is shown in Figure 1c. Lattice fringes corresponding to MAPbBr_3−x_Cl_x_ (x = 1.8) crystals are clearly. The nanocrystals with an average size of 3.41 nm are monodispersed in the PVDF nanofiber. Appendix A summarizes the size distribution of the nanocrystals. Intriguingly, the size of PQDs was elongated along the longitudinal direction of the nanofiber. This is considered to be related to the electric field force. In the PVDF solution, perovskite precursor molecules aggregate into nanoclusters. These nanoclusters were elongated by the electric field force during the electrospinning process and subsequently, with the evaporation of solvents, the molecular clusters grow into perovskite nanocrystals in situ.

The fact that the PQDs can retain several nanometers, which is beneficial for luminescence efficiency of QDs, is attributes to the interaction between MA^+^ of MACl (or MABr) and −CF_2_− of PVDF [27]. This interaction prevents MA^+^ from aggregating and thus restricts the growth of perovskite molecular clusters. In addition to passivating surface states and restricting the growth of PQDs, PVDF can also provide an environment for the long-term stability of PQDs due to its hydrophobicity and solidity. The XRD pattern of the PVDF membrane prepared by electrospinning is shown in Figure 2. An obvious diffraction peak at 20.2° is corresponded with β-phase PVDF, indicating a good crystallinity [27].

### 3.2. Effects of Halogens Composition in MAPbBr_3−x_Cl_x_ QDs

The MAPbBr_3−x_Cl_x_ (*x* = 0, 1.8, or 3) nanocrystals in the PVDF nanofiber can be identified from the XRD patterns (Figure 2). For clarity, the positions of XRD diffraction peaks and their corresponding crystal planes [33] were summarized in Table 1. These diffraction peaks are distinct evidence that the as-synthesized MAPbBr_3−x_Cl_x_ is a cubic phase. Intriguingly, by comparing the XRD patterns of MAPbBr_3_, MAPbBr_1.2_Cl_1.8_ and MAPbCl_3_, we found that with increasing the chlorine content in MAPbBr_3−x_Cl_x_ nanocrystals, the main diffraction peaks slightly shift towards the larger angle. The main reason is that the radius of chloride ion (181 pm) is smaller than that of bromine ion (195 pm), so the lattice constant of MAPbBr_3−x_Cl_x_ nanocrystals tends to decrease with the increase of chlorine content, and according to Bragg diffraction theory, the corresponding diffraction angle increases.

The effect of halogens composition in PQDs on their spectrum were scrutinized by applying the UV-vis absorption and PL emissions. The normalized light absorption spectra of the PVDF-PQDs membranes are shown in Figure 3a. The strong absorptions start at 405, 455 and 525 nm for MAPbCl_3_, MAPbBr_1.2_Cl_1.8_ and MAPbBr_3_, respectively and are blue-shifted with chlorine replacing bromine, which is associated with bandgap widening upon chlorine doping [13]. According to the Tauc plot (Figure 3b), the bandgaps of MAPbCl_3_, MAPbBr_1.2_Cl_1.8_ and MAPbBr_3_ QDs in the PVDF nanofibers are calculated to be 2.98, 2.58 and 2.28 eV, respectively, which was obtained by reading the abscissa value of the intersection of the tangent and the x-axis.

By adjusting the ratio of Cl^−^ and Br^−^, a series of PVDF-perovskites membranes of different emission wavelengths were prepared (Appendix A). Especially, Figure 3c compares PL spectra of MAPbCl_3_, MAPbBr_1.2_Cl_1.8_ and MAPbBr_3_ QDs encapsulated in PVDF membranes. Corresponding with the wavelength of the absorption onset, PL peaks of MAPbCl_3_, MAPbBr_1.2_Cl_1.8_ and MAPbBr_3_ QDs excited by 330 nm light are at 407, 456, and 527 nm and their full width at half maximum are of 67.59, 31.86, and 30.47 nm, respectively. To characterize the uniformity of PQDs distribution in the membrane, five points on the membrane were selected to compare their PL intensities (Appendix A). The PL spectra of the five points were almost coincident, which confirms that the distribution of PQDs is uniform in the membrane (Appendix A). The comparison of luminous colors of PVDF-MAPbBr_1.2_Cl_1.8_ and PVDF-MAPbBr_3_ QDs membranes are shown in Appendix A. Furthermore, in comparison to the wavelength of starting absorption, the emission peaks of each sample slightly shifted to the longer wavelength, which is known as the Stokes-shift [34]. Take MAPbCl_3_ as an example, its PL peak is at 407 nm wavelength whereas the starting absorption wavelength is at 405 nm. Because the well-crystallized PVDF is solid and compact, it is more suitable for the protection of PQDs. The PL peak value and its wavelength position of a random sample of PVDF-MAPbBr_1.2_Cl_1.8_ were monitored for 30 days under the same measuring conditions, as shown in Figure 3d. Notably, the PL attenuation is less than 1%, while the wavelength shift is within 1 nm due to the slight growth of the PQD gains, which indicates an excellent stability of the PVDF-PQDs.

### 3.3. Plasmon-Enhanced Fluorescence

In comparison to MAPbBr_3_ QDs, the luminous efficiency of MAPbBr_3−x_Cl_x_ QDs has been hovering around very low values due to the increase in the undesirable effects of defect states. To improve the luminous efficiency, we explored plasmon-enhanced fluorescence of MAPbBr_3−x_Cl_x_ by introducing PVP-Ag nanofibers into PQD nanofibers membranes. For PVP-Ag nanofibers, the Ag NPs were encapsulated by PVP which acts as a good stabilizer. Two sp orbitals of the Ag^+^ ions could be occupied by the donated lone pair electrons of both oxygen and nitrogen atoms in PVP monomers, which effectively stabilizes Ag^+^ ions and decreases their chemical potential [35]. Figure 4a and b display HRTEM images of PVP-Ag nanofiber and Ag NPs. Clearly, the Ag NPs are well-dispersed in a nanofiber without aggregation, and these particles are highly spherical with an average size 13 nm. The absorbance spectrum of Ag NPs was measured in DMF solution, which shows a strong absorption peak at 470 nm. MAPbBr_1.2_Cl_1.8_ nanocrystals were selected for plasmon-enhanced experiments owing to its PL spectrum coinciding well with the plasmonic resonance band of Ag NPs (shown in Figure 4c), which promisingly produced a high fluorescence enhancement [23,24]. PVP-Ag nanofibers and PQD nanofibers were deposited on a substrate by layer-by-layer electrospinning to form a composite membrane, where Ag nanofibers of each layer interweaved with PQD nanofibers of the adjacent layer.

We defined MAPbBr_1.2_Cl_1.8_-layer/Ag-layer/ MAPbBr_1.2_Cl_1.8_-layer as an analytical periodic unit as shown in Figure 5a. Figure 5b compares the emission spectra of periodic units with and without an Ag-layer. Clearly, in comparison with the MAPbBr_1.2_Cl_1.8_ membrane, Ag nanofibers located in the MAPbBr_1.2_Cl_1.8_ membrane can significantly enhance the PL intensity. The enhancement effect is affected by the concentration of Ag NPs in PVP nanofibers. The PL signal increases as more and more Ag NPs have been introduced in nanofibers up to a certain concentration. The maximum fluorescence intensity was enhanced to be ~4.8 folds of initial fluorescence intensity after the additive amount of Ag NPs to the optimized concentration 0.236 M. However, since Ag NPs increased beyond the optimal concentration, fluorescence intensity decreased, which is aligned with previous results [23,24]. It is believed that in addition to enhancing fluorescence, Ag NPs also act as absorbers and scatters of the fluorescence of MAPbBr_1.2_Cl_1.8_ QDs, and with the increase concentration of nanoparticles, the fluorescence loss becomes dominant. A different explanation is proposed later in this paper based on the investigation of optical field distribution of Ag nanofibers.

Moreover, in comparison with the PVDF-MAPbBr_1.2_Cl_1.8_ single layer film, the maximum emission peak of the multi-layer film shows the ~8 nm blue-shift as shown in Figure 5c. This blue shift is caused by the introduction of PVP rather than Ag NPs. To prove this, we introduced a layer of pure PVP-nanofiber film at the middle of the PVDF-MAPbBr_1.2_Cl_1.8_ membrane. Obviously, the PL peak shifts ~8 nm to the short wavelength. However, no blue or red shift of the PL peak was observed upon adding Ag NPs into the PVP-nanofiber.

### 3.4. Theoretical Analysis of the Plasmon-Enhanced Fluorescence

The plasmon-enhanced luminescence is usually caused by the local electric field intensification close to the metallic surface. For too-short distances between the emitter and the metal surface, however, non-radiative transitions are usually dominant, thereby quenching the radiative emission. The quenching is usually caused by energy losses in the metal, such as interband transitions [36]. In the proposed structure (Figure 5a), the polymers (PVDF and PVP) that are highly transparent in the emission range of MAPbBr_3−x_Cl_x_ QDs separate the emitters from Ag NPs at an appropriate distance, and therefore eliminate the fluorescence quenching. According to TEM measurements of PVDF-PQDs nanofibers and PVP-Ag nanofibers, the average nearest distance between PQDs and Ag NPs can be deduced to be 7 nm, which is within the near-field enhancement of plasmonic nanoparticles [26,36]. In addition to the near-field, it is reported that the far-field of plasmonic nanoparticles also contributes to the fluorescence enhancement [26]. In order to gain insight into the fluorescence enhancement ability of PVP-Ag nanofibers, we simulated the optical field distribution of the PVP-Ag nanofiber irradiated by the 330 nm light that excites the fluorescence of MAPbBr_1.2_Cl_1.8_ QDs. On the basis of SEM and TEM observation on as-prepared NPs, a calculation model representing PVP-Ag nanofiber containing a certain Ag NPs concentration in the PVP membrane is established. The geometric design of Ag NPs and polymer nanofibers was determined by the average measured values of as-prepared samples. In particular, Ag NPs with average size of 13 nm are distributed in a PVP nanofiber at the particle density of 30 per nanometer in the simulated model, and the diameter ratio of PVP nanofiber to PVDF nanofiber is ~5. Light scattering by PQDs with only a few nanometers is negligible, hardly effecting the optical field distribution of the PVP-Ag nanofiber [37]. Therefore, in order to highlight the main factors, PQDs are not included in the PVDF-nanofiber computational model. Figure 6c shows the optical field distribution of PVP-Ag nanofiber. Surprisingly, in comparison with the incident optical field, the strong optical field distributes within the larger region around the Ag nanofiber, which is consistent with the reported far-field light enhancement of plasmonic nanoparticles [26]. The optical field intensity at just several nanometers away from Ag NPs can be enhanced to 10^5^ times of the background optical field. This is undoubtedly one of the main reasons for the fluorescence enhancement of MAPbBr_1.2_Cl_1.8_ QDs by adding Ag nanofibers in the MAPbBr_1.2_Cl_1.8_ QD membrane.

It is noteworthy that in the model simulated above, the density of Ag NPs in PVP nanofiber is 30/nm, corresponding to the average measured density of Ag NPs in as-prepared PVP-Ag nanofibers obtained by using the optimized concentration 0.236 M of Ag NPs. When the concentration of Ag NPs increased to 0.324 M, the average measured density of Ag NPs in as-prepared PVP-Ag nanofibers is 40/nm. On the basis of this density, the optical field around Ag nanofiber was simulated again and the results are shown in Figure 6d. Intriguingly, the optical field of Ag NPs shows strong locality and most of the light energy is concentrated between adjacent Ag NPs due to plasmon resonance coupling. However, the extended range of the enhanced light field around Ag NPs decreases, indicating that the enhanced range of fluorescence reduces accordingly. Therefore, we believe this is the main reason for fluorescence intensity decreases upon increasing Ag NPs beyond the optimal concentration.

## 4. Conclusions

In summary, we have developed a cost-efficient high-throughput strategy for fabricating stable PVDF-PQDs membranes using the electrospinning method. With PVDF completely encapsulating, PQDs were excellently surface-passivated and well insulated from water and oxygen. Moreover, the molecule group –CF_2_– of PVDF prevents MA^+^ of PQDs from aggregating, thus enabling PQDs to retain several nanometers, which is beneficial for luminescence efficiency. A series of PVDF-MAPbBr_3−x_Cl_x_ membranes were fabricated. By controlling the ratio of Cl^−^ to Br^−^ of MAPbBr_3−x_Cl_x_, the position of the PL peak varies from 402 to 527 nm, so that the color of the emitting light can be tuned flexibly. To improve luminous efficiency, we successfully introduced a layer of PVP-Ag nanofibers into the PVDF-PQDs membrane via layer-by-layer electrospinning and this well-considered structure produced 4.8 folds fluorescence enhancement of the PQDs. The enhancement effect of Ag nanofibers was scrutinized by simulating the optical field distribution of PVP-Ag nanofiber. Our strategy is robust as it can be easily extended to fabricate polymer-encapsulated nanoparticle membranes of diversified materials with engineered properties for applications in LEDs, lasers, light-emitting transistors and photovoltaic devices.

## Figures and Tables

**Figure 1 nanomaterials-09-00770-f001:**
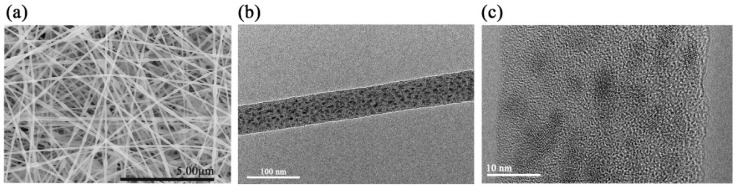
(**a**) Scanning electron microscopy (SEM) image of polyvinylidene fluoride (PVDF)-MAPbBr_1.2_Cl_1.8_ nanofibers. (**b**) Transmission electron microscope (TEM) image of a single PVDF nanofiber with embedded MAPbBr_1.2_Cl_1.8_. (**c**) High-resolution TEM (HRTEM) image of MAPbBr_1.2_Cl_1.8_ quantum dots (QDs) in PVDF nanofiber.

**Figure 2 nanomaterials-09-00770-f002:**
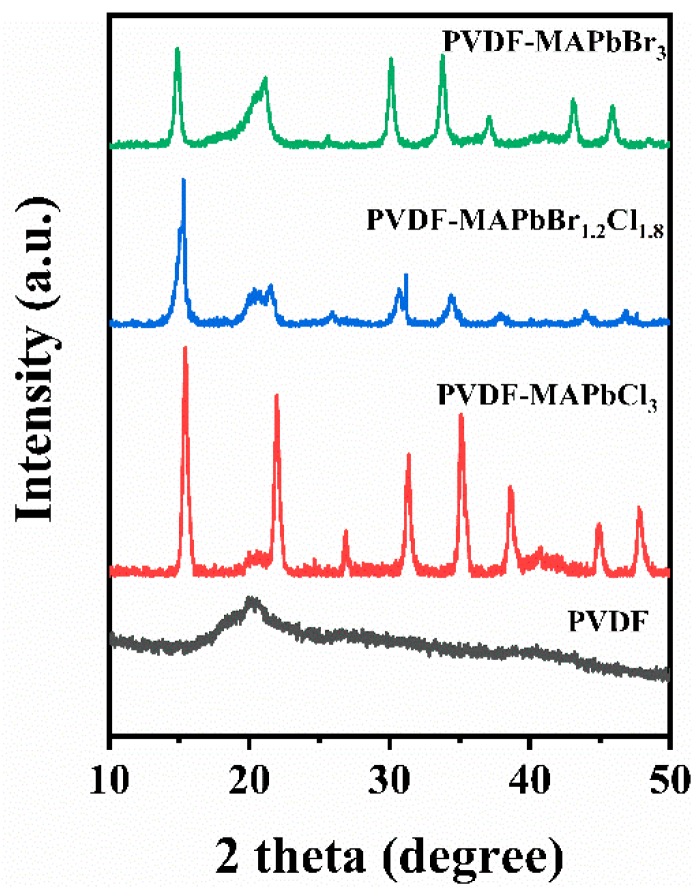
The X-ray diffraction (XRD) patterns of the PVDF-MAPbBr_3−x_Cl_x_ (x = 1.0, 1.8, 3.0) membranes and the PVDF membrane.

**Figure 3 nanomaterials-09-00770-f003:**
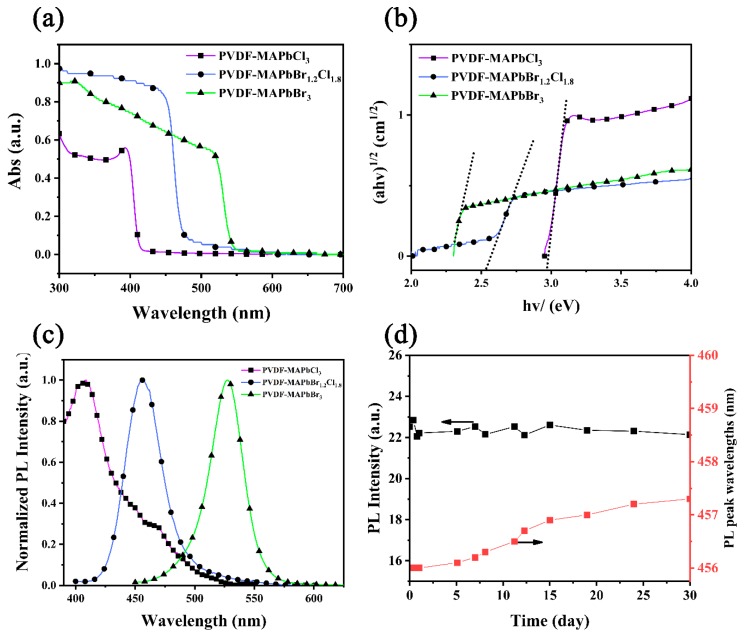
(**a**) Normalized absorption spectra, (**b**) Tauc plot and (**c**) normalized photoluminescence (PL) spectra of PVDF-perovskite quantum dots (PQDs) membranes. (**d**) Daily record data of the PL peak value and its wavelength position of the PVDF-MAPbBr_1.2_Cl_1.8_ membrane which was kept in the air at room temperature.

**Figure 4 nanomaterials-09-00770-f004:**
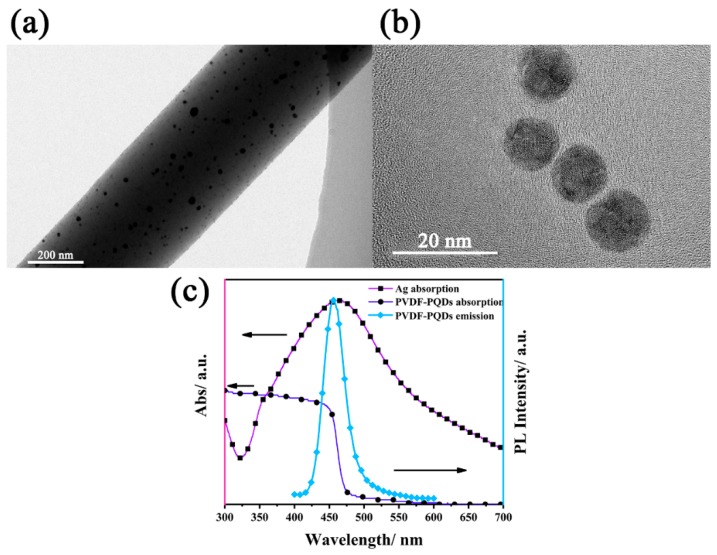
HRTEM images of the (**a**) PVP-Ag nanofiber and (**b**) silver nanoparticles (Ag NPs); (**c**) Absorbance spectra of Ag NPs and PVDF-MAPbBr_1.2_Cl_1.8_, and PL spectrum of PVDF-MAPbBr_1.2_Cl_1.8_.

**Figure 5 nanomaterials-09-00770-f005:**
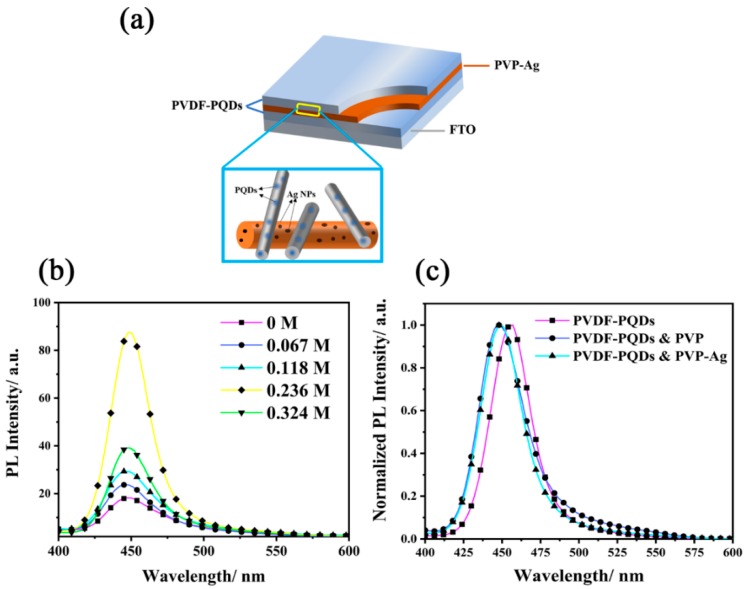
(**a**) The schematic diagram of the multilayer membrane structure. The insert is enlarged view of PVP-Ag and PVDF-PQD nanofibers on the interface between PVP-Ag and PQD layers (**b**) PL spectra of the composite membranes with a layer of PVP-Ag nanofibers sandwiched in the MAPbBr_1.2_Cl_1.8_ membranes, and the concentrations of Ag NPs varying from 0 to 0.324 M. (**c**) Normalized PL spectra of PVDF-MAPbBr_1.2_Cl_1.8_ membranes, the sandwich structural with a PVP film between the PVDF-MAPbBr_1.2_Cl_1.8_ films and the sandwich structural with PVP-Ag nanofiber film between the PVDF-MAPbBr_1.2_Cl_1.8_ films.

**Figure 6 nanomaterials-09-00770-f006:**
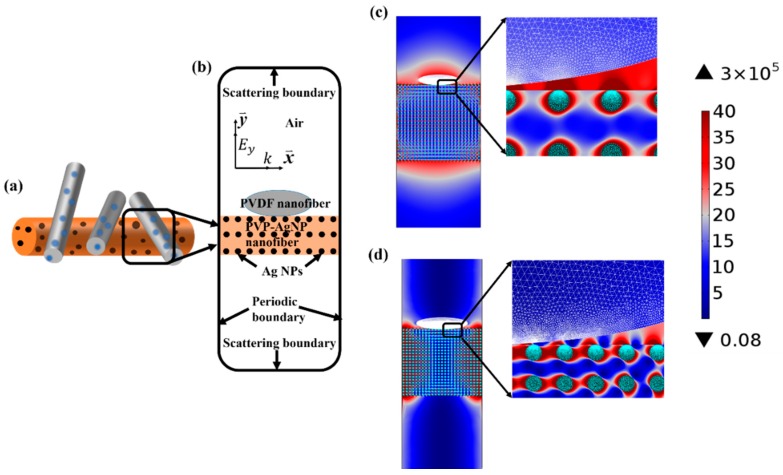
(**a**) The schematic diagram of the intertexture of PVP-Ag nanofibers and PQD nanofibers on the interface between PVP-Ag and PQD layers; (**b**) a cross-section of the computational geometry. The calculation dimension is truncated by scattering boundary conditions and periodic boundary conditions. The incident optical wave is polarized along *y*-axis, traveling along x-axis. $E_{y}$ and *k* are the incident electric field and wavevector, respectively; (**c**) and (**d**) Plots of relative optical field intensity $E^{2}=\left(E\cdot E^{*}\right)/{\left|E_{inc}\right|}^{2}$ corresponding to 0.236 and 0.324 M of Ag NPs concentration in PVP-Ag nanofibers, respectively, where $\left(E\cdot E^{*}\right)$ is the amplitude of the total optical field intensity and ${\left|E_{inc}\right|}^{2}=1\;J/{\mathrm{nm}}^{2}$ is the amplitude of the incident optical field intensity. The inserts in both (**c**) and (**d**) are enlarged view of the distribution of optical field intensity at the intersection of both PVP-Ag and PQD nanofibers.

**Table 1 nanomaterials-09-00770-t001:** XRD diffraction peaks and corresponding crystal planes of PVDF-MAPbBr_3−x_Cl_x_ membranes.

Samples	Diffraction Peaks and Crystal Planes							
MAPbBr_3_	14.82°	21.04°	25.98°	30.11°	33.72°	37.09°	43.09°	45.91°
MAPbBr_1.2_Cl_1.8_	15.30°	21.54°	26.34°	30.83°	34.37°	38.06°	44.04°	46.83°
MAPbCl_3_	15.42°	21.97°	26.89°	31.30°	35.11°	38.60°	44.93°	47.85°
Crystal planes	(001)	(011)	(111)	(002)	(021)	(211)	(022)	(003)

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
