# Peer review of "Plasmon-Enhanced Blue-Light Emission of Stable Perovskite Quantum Dot Membranes"

_nanomaterials, 2019, doi:10.3390/nano9050770_

Reviewer 1 Report

The paper entitled “Plasmon-enhanced blue-light emission…” by Kai Gu et al. presents the experimental and theoretical demonstration of enhancement of large-area Perovskite quantum dot membranes.

I have found the paper to be interesting to researchers in the field, the paper is well written and present results in right context of the current research in the field.

One remark should be made. The parameters for theoretical modeling have been taken from the paper Ref. [38] where the dielectric constants are presented in IR and far-IR spectral regions. The authors should described how they obtained the dielectric constant in the spectral region of their research. It would improve presentation of the paper. 

In conclusion, in my opinion, the paper should be accepted for publication once the authors respond to the remarks above.

Author Response

We addressed all the insightful comments and please find the modifications in the manuscript nanomaterials-498677. The changes in light of the comments are used the “Track Changes” function in the revised manuscript for your convenience.

The paper entitled “Plasmon-enhanced blue-light emission…” by Kai Gu et al. presents the experimental and theoretical demonstration of enhancement of large-area Perovskite quantum dot membranes.

I have found the paper to be interesting to researchers in the field, the paper is well written and present results in right context of the current research in the field.

One remark should be made. The parameters for theoretical modeling have been taken from the paper Ref. [32] where the dielectric constants are presented in IR and far-IR spectral regions. The authors should described how they obtained the dielectric constant in the spectral region of their research. It would improve presentation of the paper.

In conclusion, in my opinion, the paper should be accepted for publication once the authors respond to the remarks above.

Reply: We very appreciate you for these insightful comments. Previously, we reported the simulation of optical field distribution of Ag NPs. On the basis of the reported calculation technology, the simulation on PVP-Ag NPs fiber was carried out here. The Ref.[30] for Ag dielectric constants was misprinted and it was replaced by the literature “Johnson, P. B.; Christy, R.-W., Optical constants of the noble metals. Physical review B 1972, 6 (12), 4370”. The manuscripts was modified according to suggestion and please see Ref [32] in the last page of the paper.

Thank you very much.

Reviewer 2 Report

In this manuscript the authors report some kind of nanocomposite containing a polymer matrix together with perovskite nanoparticles and silver nanorods. The authors first claim the advantages of the proposed technology, and then present an enhancement of photoluminescence with the concentration of the silver nanoparticles. I have found, however, several inconsistences that prevent the publication of the paper, at least in its current form.

1. In general English has to be improved, mainly in the introduction, but also in someparts of the text. I have found also several misprints and acronyms that are not defined (PVDF for example). I recommend authors to reread carefully their paper.

2. Authors should explain what the experimental conditions in Figure 1d are. They present absolute value of PL that van be influenced by many factors (region of the sample, stability of the laser...). Do they fix the position of the spot on the sample during days? Do they perform any statistics?

3. Authors start first paragraph of page 4 by this sentence : "MAPbBr1.2Cl1.8-layer/Ag-layer/ MAPbBr1.2Cl1.8-layer as an analytical periodic unit". Then, I understand they present some kind of layer by layer technology. However, data in Figure 5 is parametrized with concentration. Then, is system composted by three layers? A scheme of the samples would be helpful.

4. Data in Figure 5 present apparently an enhancement of the PL, presented, again, by the absolute value, which could be affected, as I said by many factors. A reliable absolute value of PL can depend on the region of the sample, stability of the beam or the detector... Here, however, I understand they use different samples from one measurement to another. In this case, at least some statistic is mandatory (several points in the same sample).

5. I think the discussion in section 3.4 has to be improved. As far as I understand authors calculate the electric field distribution, and then they argue a highest near electric field with the plasmonic nanoparticle. Here I have to say:

- A high near electric field of plasmonic nanoparticles has been reported in many publications, but no reference is included.

-  Figure 6 is not completely clear. For example, I miss size of simulation windows, a simulation of the fiber without Ag, or I do not understand what is wanted to show in the zoom. In the scheme of Figure 6b there are only two layers, although I understood there are three (comment 3).

- In addition, I do not agree completely with the discussion. An enhancement of plasmonic nanoparticles can be obtained by a higher scattering cross section or by a weak coupling between the plasmon and the exciton. In the former a coupling is not mandatory, since there is just an increase or radiated light (in some cases in a preferable direction). In the later there is a plasmon-photon coupling that can lead to a quenching or enhancement of the PL depending on the thickness of the plasmonic nanoparticle and emissive material. Here, since there isn't any discussion about a spacer, I understand that authors would like to report some kind of preferable scattering in the perpendicular direction. The discussion is limited, however, by the calculation of the electric field distribution. But what does it mean? Optical modes of the metal nanoparticles? or there is an external source? How is polarization included?

I think the work is ok on a chemical point of view, but it presents several physical inconsistencies.

Author Response

We addressed all the insightful comments and please find the modifications in the manuscript nanomaterials-498677. The changes in light of the comments are used the “Track Changes” function in the revised manuscript for your convenience.

1. In general English has to be improved, mainly in the introduction, but also in some parts of the text. I have found also several misprints and acronyms that are not defined (PVDF for example). I recommend authors to reread carefully their paper.

Reply: We have revised the English expression, misprints, and acronyms of the full text, please see the changes in the “Track Changes” function used in the revised manuscript. Thanks

2. Authors should explain what the experimental conditions in Figure 1d are. They present absolute value of PL that can be influenced by many factors (region of the sample, stability of the laser...). Do they fix the position of the spot on the sample during days? Do they perform any statistics?

Reply: As-prepared samples were kept in the air at room temperature. The monitoring of the sample stability was carried out under the same measurement conditions (please see Fig.3d in the revised manuscript). To characterize the uniformity of PQDs distribution in the membrane, five points on the membrane were selected randomly to compare their PL intensities, and their PL spectra are almost coincident even after ~4 months, which confirms that the distribution of PQDs is uniform in the membrane. The manuscripts was modified according to suggestion and please see it from 223th to 226th lines under the “Track Changes” function in the revised manuscript and Figure S4 in the Supporting Information. Thanks 

3. Authors start first paragraph of page 4 by this sentence: "MAPbBr1.2Cl1.8-layer/Ag-layer/ MAPbBr1.2Cl1.8-layer as an analytical periodic unit". Then, I understand they present some kind of layer by layer technology. However, data in Figure 5 is parametrized with concentration. Then, is system composted by three layers? A scheme of the samples would be helpful.

Reply:  The schematic diagram of the samples with the detailed information was added in the revised manuscript according to the suggestion and please see Figure 5(a). Thanks

4. Data in Figure 5 present apparently an enhancement of the PL, presented, again, by the absolute value, which could be affected, as I said by many factors. A reliable absolute value of PL can depend on the region of the sample, stability of the beam or the detector... Here, however, I understand they use different samples from one measurement to another. In this case, at least some statistic is mandatory (several points in the same sample).

Reply:  The manuscripts was modified according to suggestion and please see it from 223th to 226th lines under All Markup of the “Track Changes” function in the revised manuscript and Figure S4 in the supporting information. Thanks 

5. I think the discussion in section 3.4 has to be improved. As far as I understand authors calculate the electric field distribution, and then they argue a highest near electric field with the plasmonic nanoparticle. Here I have to say:

- A high near electric field of plasmonic nanoparticles has been reported in many publications, but no reference is included.

Reply:  According to the suggestion, the relevant references have been added in the 142th and 296th lines (under All Markup of the “Track Changes” function) in the revised manuscript. Thanks

-  Figure 6 is not completely clear. For example, I miss size of simulation windows, a simulation of the fiber without Ag, or I do not understand what is wanted to show in the zoom. In the scheme of Figure 6b there are only two layers, although I understood there are three (comment 3).

Reply:  We added a detailed description of the Figure 6 according to the suggestion and please see it in the revised manuscript. Thanks.

- In addition, I do not agree completely with the discussion. An enhancement of plasmonic nanoparticles can be obtained by a higher scattering cross section or by a weak coupling between the plasmon and the exciton. In the former a coupling is not mandatory, since there is just an increase or radiated light (in some cases in a preferable direction). In the later there is a plasmon-photon coupling that can lead to a quenching or enhancement of the PL depending on the thickness of the plasmonic nanoparticle and emissive material. Here, since there isn't any discussion about a spacer, I understand that authors would like to report some kind of preferable scattering in the perpendicular direction. The discussion is limited, however, by the calculation of the electric field distribution. But what does it mean? Optical modes of the metal nanoparticles? or there is an external source? How is polarization included?

Reply:  We very appreciate you for these insightful comments. Previously, we reported the simulation of optical field distribution of Ag NPs. On the basis of the reported calculation technology, the simulation of PVP-Ag NPs fiber was carried out here.

The effects of plasmonic nanoparticles on light field distribution mainly include light scattering, plasmon-photon coupling effect and enhancement of the localized light field. Light scattering by NPs with only a few nanometers is negligible (Please see Ref [37] in the manuscript). Moreover, for the as-prepared PVDF-PQDs, almost all PQDs are encapsulated inside PVDF and the polymer separates the emitters from Ag NPs at an appropriate distance (Please see TEM and HRTEM images of PVDF-PQDs in Figure 1 in the manuscript and Figure S1 in the Supporting Information), and the light energy loss by a plasmon-photon coupling is also negligible. Therefore, the plasmon-enhanced localized optical field is a main reason for the fluorescence enhancement of PQDs.    

The schematic diagram of the samples with the detailed information (Figure 5(a)) and a detailed description of the Figure 6 have been added in the revised manuscript. The more explanations were added in the revised manuscript and please see rows 290, 276, and 311under Simple Markup of “Trank changes” function (or rows 290, 305 and 325 under All Markup of “Trank changes” function). 

Thank you very much.

Reviewer 3 Report

 The manuscript is well written and provides interesting aspects, related to membranes of polymer-encapsulated perovskite QDs fabricated by the electrospinning method. However, there are a few points that need clarifications.

1. The authors report about large area membranes but in the manuscript, the dimensions of the (at least the studied) membranes are not given. 

2. At row 154, the authors state as obvious the fact that PQDs are not embedded on the surface of the fibers, based probably on the smoothness of the surface of the fibers. Are there any other measurements to support this statement? Maybe a detailed TEM picture of a section of the fiber containing perovskite NCs?

3. Fig 1d would make more sense as fig 3d. Regarding this figure, is there an explanation for the wavelength shift, even though this shift is less than 1nm?

4. Regarding XRD measurements, it seems that the curve for PVDF-MAPbCl3 apparently needs some background extraction. Also, please explain the peaks after 40o  in Table S1 as well, since they are shown in fig 2. Additionally, peaks around 26o for PVDF-MAPbBr3 and PVDF-MAPbBr1.2Cl1.8 are not present in fig 2. Table S1 makes more sense in the manuscript, somewhere nearby fig 2, otherwise it is difficult to follow.

5. Fig 2a is not explained in the text.

6. Fig 4c, how was the abs spectra measured for AgNPs, were they in solution? If yes, what solution was that?

7. Row 248, “…with the increase of nanoparticles …” concentration most likely.

8. Fig 5a shows more like a ~4 times fluorescence enhancement

Author Response

We addressed all the insightful comments and please find the modifications in the manuscript nanomaterials-498677. The changes in light of the comments are used the “Track Changes” function in the revised manuscript for your convenience.

1. The authors report about large area membranes but in the manuscript, the dimensions of the (at least the studied) membranes are not given. 

Reply: The electrospinning method has great potential in fabricating large-area QDs films. Due to the limitation of the number of words in the paper, we haven’t given many explanations about the size of QDs films. Therefore, the “large-area” in title of the manuscript was deleted according to the suggestion. Thank you very much.

2. At row 154, the authors state as obvious the fact that PQDs are not embedded on the surface of the fibers, based probably on the smoothness of the surface of the fibers. Are there any other measurements to support this statement? Maybe a detailed TEM picture of a section of the fiber containing perovskite NCs?

Reply: More TEM and HRTEM images of PVDF-PQDs was provided according to suggestion and please see Figure S1 in the Supporting Information. Thanks

3. Fig 1d would make more sense as fig 3d. Regarding this figure, is there an explanation for the wavelength shift, even though this shift is less than 1nm?

Reply: The 1 nm wavelength shift (in Fig 3d) is due to the slight growth of PQDs.  The manuscript was modified according to the suggestion and please the changes in the “Track Changes” function used in the revised manuscript. Thanks 

4. Regarding XRD measurements, it seems that the curve for PVDF-MAPbCl3 apparently needs some background extraction. Also, please explain the peaks after 40o  in Table S1 as well, since they are shown in fig 2. Additionally, peaks around 26o for PVDF-MAPbBr3 and PVDF-MAPbBr1.2Cl1.8 are not present in fig 2. Table S1 makes more sense in the manuscript, somewhere nearby fig 2, otherwise it is difficult to follow.

Reply: The background extraction was done for all of XRD curves in Figure 2 and peaks around 26o for PVDF-MAPbBr3 and PVDF-MAPbBr1.2Cl1.8 are presented. Table S1 was transferred from the Supporting Information to the manuscript. The peaks after 40o is explained in Table 1. Please see Figure 2, Table 1. Thanks 

5. Fig 3a is not explained in the text.

Reply: The explanation of Fig 3a is added according to the suggestion and please see row 198 under Simple Markup in “Trank changes” function (or row 211 under All Markup of “Trank changes” ) in the revised manuscript. Thanks 

6. Fig 4c, how was the abs spectra measured for AgNPs, were they in solution? If yes, what solution was that?

Reply: The absorbance spectrum of Ag NPs was measured in DMF solution. The manuscripts was modified according to the suggestion and please see row 237 under Simple Markup in “Trank changes” function (or row 250 under All Markup in “Trank changes” ) in the revised manuscript. Thanks 

7. Row 248, “…with the increase of nanoparticles …” concentration most likely.

Reply: The manuscripts was modified according to the suggestion and please see row 257 under Simple Markup in “Trank changes” function (or row 271 under All Markup in “Trank changes” ) in the revised manuscript. Thanks 

8. Fig 5a shows more like a ~4 times fluorescence enhancement

Reply: We reanalyzed the fluorescence enhancement. A 4.8 times fluorescence enhancement was obtained. The manuscripts was modified according to the suggestion and please see the changes in the “Track Changes” function used in the revised manuscript. Thanks 

Thank you very much.

Round  2

Reviewer 2 Report

Now, authors have addressed all my comments and I think the manuscript is ready for publication. I would only recommend authors to include the thickness of the spacer at the beginning of section 3.4 and a reference supporting that this thickness gives rise to an enhancement. I have also found the following minor English problems:

Line 31 replace become by has become

Line 81 replace in other papers by reported elsewhere

Line 153. As an example by For example

Author Response

Thank you for the insightful comments. According to TEM measurements of PVDF-PQDs nanofibers and PVP-Ag nanofibers, the average nearest distance between PQDs and Ag NPs can be deduced to be 7-nm, which is within the near-field enhancement of plasmonic nanoparticles.(Please see Ref.26)  In addition to the near-field, it is reported that the far-field of plasmonic nanoparticles also contributes to the fluorescence enhancement. The simulation of both near- or far-field distributions of the PVP-Ag nanofiber is consistent with the reported. (Please see Ref.26) 

The manuscripts was modified according to suggestion and please see row 3, 83, 153 and 282 under Simple Markup in “Trank changes” function in the revised manuscript.

Thank you very much.